# Gas6 Prevents Epithelial-Mesenchymal Transition in Alveolar Epithelial Cells via Production of PGE_2_, PGD_2_ and Their Receptors

**DOI:** 10.3390/cells8070643

**Published:** 2019-06-26

**Authors:** Jihye Jung, Ye-Ji Lee, Youn-Hee Choi, Eun-Mi Park, Hee-Sun Kim, Jihee L. Kang

**Affiliations:** 1Department of Physiology, School of Medicine, Ewha Womans University, Seoul 07804, Korea; 2Tissue Injury Defense Research Center, School of Medicine, Ewha Womans University, Seoul 07804, Korea; 3Department of Pharmacology, School of Medicine, Ewha Womans University, Seoul 07804, Korea; 4Department of Molecular Medicine, School of Medicine, Ewha Womans University, Seoul 07804, Korea

**Keywords:** Gas6, prostaglandins, epithelial-mesenchymal transition, alveolar epithelial cells

## Abstract

The epithelial-mesenchymal transition (EMT) is important in organ fibrosis. We hypothesized that growth arrest-specific protein 6 (Gas6) and its underlying mechanisms play roles in the prevention of EMT in alveolar epithelial cells (ECs). In this study, to determine whether Gas6 prevents TGF-β1-induced EMT in LA-4 and primary alveolar type II ECs, real-time PCR and immunoblotting in cell lysates and ELISA in culture supernatants were performed. Migration and invasion assays were performed using Transwell chambers. Pretreatment of ECs with Gas6 inhibited TGF-β1-induced EMT based on cell morphology, changes in EMT marker expression, and induction of EMT-activating transcription factors. Gas6 enhanced the levels of cyclooxygenase-2 (COX-2)-derived prostaglandin E_2_ (PGE_2_) and PGD_2_ as well as of their receptors. COX-2 inhibitors and antagonists of PGE_2_ and PGD_2_ receptors reversed the inhibition of TGF-β1-induced EMT, migration, and invasion by Gas6. Moreover, knockdown of Axl or Mer reversed the enhancement of PGE_2_ and PGD_2_ and suppression of EMT, migration and invasion by Gas6. Our data suggest Gas6-Axl or -Mer signalling events may reprogram ECs to resist EMT via the production of PGE_2_, PGD_2_, and their receptors.

## 1. Introduction

The epithelial to mesenchymal transition (EMT) events have been very well defined during embryogenesis, organ development, wound healing and stem cell behaviors [1]. This process has also been shown to contribute pathologically to organ fibrosis and cancer progression. Emerging evidence suggests that EMT is an important event in idiopathic pulmonary fibrosis (IPF) [2,3,4]. IPF likely results from recurrent alveolar epithelial cell (EC) injury coupled with accumulation and differentiation of fibroblasts into myofibroblasts, leading to the accumulation of extracellular matrix (ECM) and destruction of the lung parenchyma [5]. Alveolar ECs undergoing epithelial-mesenchymal transition (EMT), a process frequently mediated by TGF-β [6], are a major source of fibroblasts [7,8]. Although several drugs are currently used to treat IPF, no proven, efficacious therapies currently exist [9].

Growth arrest-specific protein 6 (Gas6) is a secreted vitamin K-dependent protein. Gas6 comprises an N-terminal γ-carboxyglutamic acid (Gla) domain followed by four epidermal growth factor (EGF)-like domains and a large C-terminal region homologous to the sex hormone binding globulin (SHBG) [10]. Gas6 is expressed in lung, heart, kidney, intestine, fibroblasts, endothelial cells, bone marrow cells [11], vascular smooth muscle [12], leukocytes [13], and neurons [14]. It is a common ligand of the Tyro3/Axl/Mer (TAM) receptor subfamily [15]. These receptors share significant domain similarity all containing two extracellular N-terminal immunoglobulin-like domains as well as two fibronectin-III-like domains followed by a tyrosine kinase domain that lies at the C-terminal cytoplasmic end of the receptors [15]. TAM activation by Gas6 mediates cell survival, proliferation, phagocytosis, differentiation, adhesion, migration, platelet function, and thrombus stabilization [11,12,13,14,15,16,17,18,19,20]. Rothlin and colleagues reported that the engagement of Gas6/TAM signaling after Toll-like receptors (TLR) activation usurps the type 1 interferon receptor (IFNAR)-STAT1 cassette to induce TLR suppressors, suppressor of cytokine signaling 1 (SOCS1) and SOCS3, resulting in the pleiotropic inhibition of both cytokine receptor and TLR signaling pathways in dendritic cells [21]. Recent study demonstrated that the Mer pathway involving phosphatidylinositol 3-kinase/Akt and nuclear factor-kappaB (NF-κB) in macrophages is responsible for decreases in pro-inflammatory signals in response to LPS or zymosan independent of new protein synthesis [22,23]. Our previous study demonstrated that Gas6/Mer/Akt/LXR signaling in macrophages modulates TLR-induced cytokine secretion, regulating the immune response [24]. These results highlight the importance of the TAM signaling for modulation of the innate immune response. In addition, we reported that macrophages can be reprogrammed by Gas6 to promote EC proliferation and wound repair via HGF, which is induced by the Mer pathway in macrophages [25]. Gas6/Axl signaling plays an important role in the control of lens EC growth and survival [26]. These data suggest a key homeostatic role of Gas6 in the maintenance of tissue ECs. However, a direct role for Gas6 in EC homeostasis through regulating EMT remains unknown.

TGF-β signaling has been shown to play an important role in EMT and development of pulmonary fibrosis [6,27]. In the induction of EMT, the activated Smad or non-Smad signaling mediates transcriptional regulation through the well-orchestrated actions of the Snail, Zeb, and Twist transcription factors [28], resulting in repression of epithelial marker gene expression and activation of mesenchymal gene expression. TGF-β1-induced EMT is an important step implicated in epithelial cell migration and invasion toward the interstitial area and the alveolar space for progression of lung fibrosis [1]. Based on these findings, here, we identified a novel role for Gas6 in the prevention of TGF-β1-induced EMT and signalling pathway as well as migration and invasion of murine alveolar type II-like lung (LA-4) ECs and primary mouse lung alveolar type II (ATII) ECs. Because cyclooxygenase-2 (COX-2)-derived prostaglandin E2 (PGE_2_) and PGD_2_ were found to negatively modulate fibrotic remodelling-associated EMT in lung ECs [29], we further investigated whether and how these factors are induced by the Gas6-Axl or -Mer pathway and mediate the inhibition of EMT, migration and invasion of ATII ECs by Gas6.

## 2. Materials and Methods

### 2.1. Reagents

Recombinant mouse Gas6 was purchased from R&D Systems (Minneapolis, MN, USA). NS-398, AH-23848, BW-A868C, BAY-u3405, PGE_2_, and PGD_2_, were purchased from Cayman Chemical (Ann Arbor, MI, USA). AH-6809, and Y-27632, (Sigma-Aldrich Chemical Co., St. Louis, MO, USA) were used as supplied. The gene-specific relative RT-PCR kit was obtained from Invitrogen (Carlsbad, CA, USA), and M-MLV reverse transcriptase was purchased from Enzynomics (Hanam, Korea). The enzyme immunoassay (EIA) kits for PGE_2_ and PGD_2_ were obtained from Assay Designs (Ann Arbor, MI, USA). The antibodies used in this study were against E-cadherin, α-SMA (Abcam, Cambrige, MA, USA), N-cadherin, c-Met, phospho-Smad2, phospho-Smad3 (Cell Signaling Technology, Beverly, MA, USA), Smad2/3 (BD Biosciences, Bedford, MA, USA), COX-1, COX-2, EP2, EP4, DP1, DP2, (Cayman Chemical), phosphor-ERK1/2, ERK, phospho-Akt, Akt, phospho-p38 MAP kinase, p38 MAP kinase (Santa Cruz Biotechnology, Santa Cruz, CA, USA), and β-actin (Sigma-Aldrich).

### 2.2. Culture of Cell Lines

LA-4, A549, and HEK-293 cells were purchased from the ATCC (Manassas, VA, USA). LA-4 and A549 cells were grown in F12K medium (Lonza, Basel, Switzerland) containing 15% heat-inactivated foetal bovine serum (FBS) at 37 °C in 5% CO_2_. HEK-293 cells were grown in Dulbecco’s modified Eagle’s medium (DMEM, Media Tech Inc., Washington, DC, USA) supplemented with 10% FBS, 2 mM l-glutamine, 100 U/mL penicillin, and 100 μg/mL streptomycin at 37 °C in 5% CO_2_.

### 2.3. Incubation of ECs

ECs were plated in six-well culture plates (2 × 10^5^ cells/well) and cultured overnight in 200 μL RPMI 1640 or DMEM containing 10% FBS. Primary ATII ECs were plated and cultured on type 1 collagen-coated culture plates (1 × 10^6^ cells/well) for 48 h. Cells were pretreated for 20 h with 400 ng/mL Gas6 before treatment with 10 ng/mL TGF-β1 (R&D Systems Inc., Minneapolis, MN, USA) [20,21]. In some experiments, 10 μM NS-398 was used to inhibit COX-2. The specific inhibitor was added 1 h before Gas6 treatment. In addition, 10 μM AH-6809, AH-23848, BW-A868C, or BAY-u3405 were used to antagonize E-prostanoid-2 receptor (EP2), EP4, DP1, or DP2, respectively. These receptor antagonists were added 1 h before the addition of TGF-β1.

### 2.4. Isolation and Culture of Primary AT II Cells

Primary murine ATII ECs were isolated from C57BL/6 mice and purified as previously described [30]. Lungs were perfused with 0.9% saline administered through the pulmonary artery until the lungs became free of blood. After lavage of lungs with 1 mL saline, dispase (100 units) was infused into cleared mice lungs, followed by incubation of the lungs for 45 min at room temperature. Thereafter, the lung tissue was separated from large bronchi by mechanical means, and the tissues were then placed to a Petri dish containing Dulbecco’s modified Eagle’s medium (DMEM) with 0.01% DNase I, in which they were incubated for 10 min at 37 °C. The cells were filtered, centrifuged, and resuspended to perform sequential plating on mouse IgG (0.75 mg/mL)–coated Petri dishes followed by cell culture dishes, each at 37 °C for 1 h, to get rid of macrophages and fibroblasts, respectively. The final cell isolates were placed in Type I collagen-coated dishes in Ham’s F12 culture medium supplemented with 15 mM C_8_H_18_N_2_O_4_S (HEPES), 0.8 mM CaCl_2_, 0.25% BSA, 5 mg/mL insulin, 5 mg/mL transferrin, 5 ng/mL sodium selenite, and 2% mouse serum. We previously reported representative confocal microscopic images of purified ATII cells (LSM5 PASCAL; Carl Zeiss, Jena, Germany) [31].

### 2.5. Transient Transfections

LA-4 cells were transiently transfected with 1 μg/mL of siRNA specifically targeting either COX-2, Axl, Mer or with control siRNA (Bioneer, Seoul, Korea) using 5 μL of siRNA transfection reagent (Genlantis, San Diego, CA, USA) according to the manufacturer’s protocol. The sequences used for COX-2 knockdown were sense 5′-CUAUGAUAGGAG CAUGUAA-3′ and antisense, 5′-UUACAU GCUCCUAUCAUA G-3′. The sequences used for Axl knockdown were sense 5′-GAGAUGGACA GAUCCUAGA-3′ and antisense, 5′-UCUAGGAUCUGUCCAUCUC-3′. The sequences used for Mer knockdown were sense 5′-CACAGUUUUAUCCUGAUGA-3′ and antisense 5′-UCAUCAGGAUAA AACUGUG-3′. The sequences for control siRNA were sense 5′-CCUACGCCACCAAUUUCG U-3′ and antisense 5′-ACGAAAUUGGUGGCGUAG G-3′. Cells were incubated in serum-free medium for 6 h for COX-2 siRNA, or 48 h for Axl and Mer siRNA prior to experimentation. None of the siRNAs used had any significant effect on cell viability.

### 2.6. Immunoblot Analysis

ECs were lysed in 0.5% Triton X-100-containing lysis buffer and resolved on a 10% SDS-PAGE gel. Separated proteins were electrophoretically transferred onto nitrocellulose and blocked for 1 h at room temperature with Tris-buffered saline containing 3% BSA. The membranes were then incubated at room temperature for 1 h with primary antibodies at 4 °C overnight and incubated with HRP-conjugated secondary antibodies for 1 h at room temperature. Proteins were visualized using enhanced chemiluminescence, according to the manufacturer’s instructions.

### 2.7. Quantitative Real-Time PCR (qPCR)

Gene expression was analyzed by real-time qPCR on a StepOnePlus system (Applied Biosystems, Life Technologies, Carlsbad, CA, USA). For each qPCR assay, a total of 50 ng cDNA was used. Primer sets for PCR-based amplifications were designed using Primer Express software. The primers used were as follows (name: forward primer, reverse primer). For mice, *Cdh1*: 5′-GCACTCTTCTCCTGGTCCTG-3′, 5′-TATGAGGCTGTGGGTTCCTC-3′; *Cdh2*: 5′-CCTCCAGAGTTT ACTGCCATGAC-3′, 5′-CCACCACTGATTCTGTATGCCG-3′; *α-SMA*: 5′-CCACCGCAAATGCTT CTAAGT-3′, 5′-GGCAGGAATGATTTGGAAAGG-3′; *Snai1*: 5′-CCCAAGGCCGTAGAGCTGA-3′, 5′-GCTTTTGCCACTGTCCTCATC-3′; *Snai2*: 5′-ATCCTCACCTCGGGAGCATA-3′, 5′-TGCCGACGAT GTCCATACAG-3′; *Zeb1*: 5′-ATTCAGCTACTGTGAGCCCTGC-3′, 5′-CATTCTGGTCCTCCACAG TGGA-3′; *Zeb2*: 5′-GCAGTGAGCATCGAAGAGTACC-3′, 5′-GGCAAAAGCATCTGGAGTTCCAG-3′; *Twist1*: 5′-TCGACTTCCTGTACCAGGTCCT-3′, 5′-CCATCTTGGAGTCCAGCTCG-3′; *Cox1*: 5′-CGATCTGGCTTCGTGAAC-3′, 5′-GAGCTGCAGGAAATAGCC-3′; *Cox2*: 5′-GGGAGTCTGGAAC ATTGTGA-3′, 5′-GTGCACATTGTAAGTAGG TG-3′; *Ptger2*: 5′-GTGGCCCTGGCTCCCGAAAGT-3′, 5′-GGCAAGGAGCATATGGCGAAGGTG-3′; *Ptger4*: 5′-ATCTTCGGGGTGGTGGGCAA-3′, 5′-CGCTTGTCCACGTAGTGGCT-3′; *Dp1*: 5′-TTTGGGAAGTTCGTGCAGTACT-3′, 5′-GCCATGAGG CTGGAGTAG A-3′; *Dp2*: 5′-CATGTGCTACTACAACTTGC-3′, 5′-GCAGACTGAAGATGTGG TAGG-3′; and *Hprt*: 5′-CCAGTGTCAATTATATCTTCAAC-3′, 5′-CAGACTGAAGAGCTACT GTAATG-3′. The cDNA abundances were normalized to that of hypoxanthine-guanine phosphoribosyltransferase (*Hprt*) and are presented as the fold-change in abundance compared to the appropriate controls.

### 2.8. Migration and Invasion Assays

Cell migration and invasion were tested using Transwell chambers (Corning Inc., Corning, NY, USA) coated with 10 μg/mL fibronectin and 300 μg/mL Matrigel matrix according to the manufacturer’s instruction, respectively. In brief, pre-incubated primary mouse AT II cells or LA-4 cells (5 × 10^4^ cells/well for the migration assay and 5 × 10^4^ cells/well for the invasion assay) in the absence or presence of TGF-β1 (10 ng/mL) were plated in replicate wells in serum-free RPMI in the upper chambers and in RPMI 1640 supplemented with 10% FBS placed in the bottom wells at 37 °C for 48 h migration time or 48 h invasion time. The nonmigrated or noninvaded cells on the upper surface of the membrane were removed with a cotton swap. Migrated cells on the lower surface were fixed with 4% paraformaldehyde and stained using 0.1% crystal violet. Three random microscopic fields (10× magnification) were photographed and counted. All experiments were performed in triplicates.

### 2.9. Enzyme-Linked Immunosorbent Assay (ELISA)

Culture supernatants were collected. The levels of PGE_2_ and PGD_2_ concentration were measured using an EIA kit according to the manufacturer’s instructions.

### 2.10. Statistical Analysis

Data are expressed as the mean ± S.E. Analysis of variance was used for multiple comparisons, and Tukey’s post hoc test was used where appropriate. The Student’s *t* test was used to compare two sample means. A *P* value less than 0.05 was considered statistically significant. All data were analysed using JMP software (SAS Institute, Cary, NC, USA).

## 3. Results

### 3.1. Gas6 Inhibits TGF-β1-Induced EMT in Lung and Kidney ECs

Pretreatment with 400 ng/mL Gas6 prevented a spindle-like morphology (Figure 1A) and changes in EMT markers, such as decreased E-cadherin and increased N-cadherin, and α-SMA, at both the protein and mRNA levels after a 48- or 72-h stimulation with TGF-β1 in LA-4 ECs (Figure 1B,C). We also observed this inhibitory effect in ATII ECs (Figure 1B,C), A549 human non-small lung cancer cells, and HEK293 human kidney cells (Appendix A). However, EMT marker protein expression was not inhibited when pretreatment occurred 2 h before TGF-β1 treatment or the culture medium was replaced 20 h after Gas6 pretreatment prior to TGF-β1 stimulation for 72 h (Appendix A).

### 3.2. Gas6 Inhibits Non-Smad TGF-β1 Signalling and EMT-Regulating Transcription Factor Expression

Gas6 pretreatment inhibited the TGF-β1-induced mRNA expression of Snai1/2, Zeb1/2, and Twist1 in LA-4 ECs, ATII ECs (Figure 2A,B), A549 cells, and HEK293 cells (Appendix A). The TGF-β1-induced increases in Snail1 and Zeb1 expression at the protein level in LA-4 cells were also reduced by Gas6 (Figure 2C). In addition, Gas6 pretreatment of LA-4 ECs did not affect the TGF-β1-induced phosphorylation of Smad2 or Smad3 (Appendix A). However, Gas6 partially inhibited the TGF-β1-induced phosphorylation of extracellular signal-regulated kinase (ERK)1/2 and Akt (Figure 2D), but not p38 mitogen-activated protein kinase phosphorylation (Appendix A).

### 3.3. Gas6 Enhances COX-2-Derived Production of PGE_2_, PGD_2_, and Their Receptors

COX-2 mRNA abundance peaked at 1 h and returned to resting levels 20 h after Gas6 treatment in LA-4 and ATII ECs (Figure 3A). COX-2 protein expression in LA-4 ECs increased up to 24 h in LA-4 ECs (Figure 3B). PGE_2_ and PGD_2_ production increased in LA-4 ECs 20 h after Gas6 treatment (Figure 3C) but was blocked by COX-2 siRNA (Figure 3D). Interestingly, mRNA and protein levels of EP2 and DP2 were enhanced 20–24 h after Gas6 treatment, whereas EP4 and DP1 mRNA and protein levels were unaffected, in LA-4 ECs (Figure 3E,F). However, transfection of LA-4 ECs with COX-2 siRNAs inhibited Gas6-indued *Ptger2* and *Dp2* mRNA expression (Figure 3G). Increases in *Ptger2* and *Dp2* mRNA expression by Gas6 were also shown in ATII ECs (Figure 3H).

### 3.4. COX-2-Derived PGE_2_ and PGD_2_ Secretion by Gas6 Treatment Inhibits EMT

The COX-2 inhibitor NS-398 reversed the effect of Gas6 on TGF-β1-induced changes in morphology (Figure 4A) and EMT markers at the protein and gene levels as well as *Snai1*, *Zeb1,* and *Twist1* mRNA expression in LA-4 (Figure 4B; Appendix A) and ATII ECs (Figure 4C,D). Moreover, knockdown of the *Cox2* gene resulted in similar effects in LA-4 ECs (Figure 4E; Appendix A). COX-2 siRNAs reversed the reduction of TGF-β1-induced phosphorylation of ERK1/2 and Akt in LA-4 ECs by Gas6 (Figure 4F).

To determine whether the production of PGE_2_ and PGD_2_ mediates anti-EMT effects, PGE_2_- or PGD_2_-specific receptor antagonists [antagonists of EP2 (AH-6809), EP4 (AH-23848), DP1 (BW-A868C), or DP2 (BAY-u3405)] were added to LA-4 and ATII ECs 1 h before TGF-β treatment in the absence or presence of Gas6. AH-6809 and BAY-u3405 significantly reversed anti-EMT effects (Figure 5A,B; Appendix A), whereas AH-23848 and BW-A868C had few effects. In addition, the reversing effects of AH-6809 and BAY-u3405 were also found in ATII ECs (Figure 5C,D).

### 3.5. Axl and Mer Receptor Tyrosine Kinases are Involved in Anti-EMT Effects of Gas6 in LA-4 ECs

Axl and Mer receptors were rapidly activated in LA-4 ECs after Gas6 stimulation (Figure 6A,B). Knockdown of Axl or Mer reversed the enhanced induction of COX-2 mRNA expression by Gas6 as well as PGE_2_ and PGD_2_ production, and *Ptger2* and *Dp2*, but not *Ptger4* and *Dp1*, mRNA expression (Figure 6C–G).

The inhibitory effects of Gas6 on EMT marker changes in LA-4 ECs at the protein and gene levels, and the downregulation of the mRNA expression of EMT transcription factors, were reversed by their specific siRNAs (Figure 6H,I; Appendix A). Axl- or Mer-specific siRNAs also reversed the reduction of TGF-β1-induced phosphorylation of ERK1/2 and Akt by Gas6 (Figure 6J; Appendix A).

### 3.6. Gas6 Inhibits TGF-β1-Induced Migration and Invasion in ATII Cells

Pretreatment of LA-4 or ATII ECs with Gas6 inhibited TGF-β1-induced cell migration and invasion (Figure 7A–D; Appendix A). However, COX-2 inhibitor NS398 or antagonists of EP2, and DP2, but not DP4 and DP1, reversed Gas6-induced inhibition of cells migration and invasion. Similarly, siRNAs of Axl or Mer also reversed the inhibitory effects of Gas6 on TGF-β1-induced migration and invasion of LA-4 ECs (Figure 7E,F).

## 4. Discussion

Gas6/Axl signaling plays an important role in the control of epithelial cell growth and survival [32]. For example, Gas6/Axl signaling mediates a survival or anti-apoptotic response in quiescent lens ECs under conditions of growth factor deprivation and proliferating lens ECs from TGFβ1- or TNFα-induced apoptosis [26]. Recently, it was proposed that Gas6/Axl signaling plays a critical role in oral epithelial cells as a key immunological regulator of host-commensal interactions [32]. These data suggest a key homeostatic role of Gas6 in the maintenance of tissue epithelial cells. In the present study, we focused on a new role for Gas6 in preventing EMT, cell migration and invasion in lung ATII ECs. Gas6 treatment 20 h prior to TGF-β1 inhibited changes in EMT markers in lung ECs, including LA-4 ECs, primary ATII ECs, A549 cells, and HEK-293 kidney cells. The master regulators of EMT, including Snail, Zeb, and Twist, are upregulated by TGF-β via both Smad-dependent and -independent mechanisms [33]. We demonstrated that Gas6 blocked non-Smad TGF-β1 signalling and downregulated the mRNA expression of EMT transcription factors. Notably, the anti-EMT effect of Gas6 was not observed after the replacement of fresh culture media prior to TGF-β addition. These data strongly suggest that Gas6 inhibits EMT through the induction and secretion of bioactive mediators to block TGF-β signalling in an autocrine/paracrine manner.

The COX-2/PGE_2_ and PGD_2_ pathways result in the inhibition of EMT in lung and renal cells [25,34]. Thus, we hypothesized that PGE_2_ and PGD_2_ secretion by Gas6 mediates anti-EMT effects in an autocrine/paracrine manner. We found that Gas6 treatment enhances COX-2 expression with secretion of these mediators from LA-4 ECs. We confirmed, using specific siRNAs for COX-2, the secretion of PGE_2_ and PGD_2_ after Gas6 stimulation. Moreover, either knockdown of COX-2 or pharmacologic inhibition of COX-2 activity by NS398 reversed the inhibition of TGF-β1-induced EMT and EMT transcription factor expression in LA-4 and ATII ECs by Gas6. In particular, COX-2 siRNAs also reversed the inhibitory effect of Gas6 on non-Smad TGF-β1 signaling in LA-4 ECs. Therefore, these data indicate that Gas6 prevents non-Smad TGF-β signaling and EMT in ATII ECs via PGE_2_ and PGD_2_ secretion.

Considering the effects of the agonists of EP1–4, the suppressive effects of PGE_2_ on TGF-β1-induced EMT may be mediated via the EP2 and EP4 receptors in A549 cells [35]. In the current study, we found that antagonists of EP2 and DP2, but not of EP4 and DP1, reversed Gas6-induced anti-EMT effects in LA-4 and ATII ECs. Interestingly, the mRNA and protein expression levels of EP2 and DP2, but not of EP4 and DP1, were enhanced up to 24 h in a time-dependent manner by Gas6 in LA-4 ECs. Taken together, these data suggest that the increases in EP2 and DP2 abundance and activity contribute, at least in part, to the anti-EMT effect of Gas6 in ATII ECs.

The functions of Gas6 appear limited to those caused by the activation of TAM receptors (Axl>Tyro3>>>Mer). In the present study, Gas6 induced higher levels of Axl phosphorylation than Mer phosphorylation in LA4 ECs. However, siRNA knockdown of Axl or Mer reversed the inhibition of changes in TGF-β1-induced EMT markers at the gene and protein levels by Gas6, restoring *Snai1, Zeb1,* and *Twist1* mRNA expression to similar levels. Of note, we found that PGE_2_ and PGD_2_ production as well as reduction in transforming growth factor (TGF)-β1-induced ERK1/2 and Akt phosphorylation levels by Gas6 were blocked by specific siRNAs for Axl or Mer. These data suggest that the Gas6-Axl and -Mer pathways mediate PGE_2_ and PGD_2_ production, maintain EC homeostasis, and prevent pathologic process EMT in LA-4 ECs after TGF-β1 stimulation. There may be an interdependence of these related receptors in LA-4 ECs. In platelets, all three TAM family receptors are expressed; the targeted knockout of a single receptor was shown to result in a phenotype of decreased platelet aggregation and clot instability [36,37]. In addition, the inhibition of related receptor tyrosine kinases was found to cause a significant decrease in astrocytoma survival and growth [38]. Further studies are needed to fully determine how receptor tyrosine kinases interact under specific biological conditions.

EMT is characterized by the enhancement of cellular migration and invasive potential. Fibroblasts and myofibroblasts from IPF patients have been shown to have distinct properties [39], including the ability to invade the ECM similar to metastatic cancer cells [40]. In addition to anti-EMT effects, we demonstrated that Gas6/Axl or Mer signaling inhibits TGF-β1-induced migration and invasion of LA-4 and ATII ECs via PGE_2_ and PGD_2_ production.

Lee et al. [41] reported that Gas6 enhanced the migration and invasion of various hepatocellular carcinoma (HCC) cell lines in the absence of TGF-β1. The Gas6-Axl pathway enhanced Snai1 mRNA expression. However, we demonstrated that Gas6 itself did not affect mRNA or protein expression of EMT markers or transcription factors in a variety of ECs; Gas6 also did not affect the migration or invasion of LA-4 or ATII ECs in the absence of TGF-β1. We believe that the diversification of Gas6 function via Mer and Axl activation may be due to differences in the experimental design of our study and those of other studies.

Gas6/Axl signaling induces proliferation of mesangial cells in kidney fibrosis [42], vascular smooth muscle cells response to intimal vascular injury [43], and cardiac fibroblasts during prolonged serum starvation. [44]. Thus, this pathway has been implicated in growth and survival processes during regeneration, and tissue repair. The selective targeting of Gas6-Axl specific antibodies or small-molecule inhibitors of TAM receptors was shown to modulate the activation of fibroblasts in patients with IPF [45]. On the other hand, an exogenous treatment with protein S, a TAM receptor ligand, significantly decreased the levels of inflammatory and profibrotic markers, decreased lung fibrosis, and blocked apoptosis in alveolar ECs [46]. These controversial data suggest that the roles of Gas6 and TAM receptors are complex; thus, future studies are necessary to more fully understand the diverse roles of Gas6-TAM signaling in certain cell types and/or in vivo disease models, considering different cellular microenvironments.

TAM receptors have been implicated in human cancers by virtue of their pathological overexpression [47,48,49,50]. Gas6 is also concomitantly overexpressed in human cancers, implying that, besides receptor overexpression, TAM participate in autocrine circuits by overexpressing both receptors and ligands [51,52,53]. Profound experimental evidence supports the unanimity that TAM receptors, in a cell-autonomous manner, act as pro-oncogenes enhancing the growth, survival, migration, and EMT of cancer cells [54]. Paradoxically, experimental evidence has demonstrated that the marked susceptibility to experimentally induced colitis observed in mutant mice lacking Gas6 as well as Axl and Mer correlates with increased incidence of colon cancer, resulting in larger tumors and reduced overall survival [55,56]. These studies indicate the complicated implication of Gas6 and TAM in cancer and underscore the importance of understanding their tissue and cell type-specific functions in cancer. To end this, further mechanistic insights into TAM-mediated regulation of anti-tumor immune responses are needed.

## 5. Conclusions

In summary, our study reveals a critical role for Gas6-Axl or -Mer signaling in the prevention of TGF-β1-induced EMT, migration, and invasion in ATII ECs. The production of COX-2-derived PGE_2_ and PGD_2_ and their receptors is essential for the effects of Gas6. Considering that EMT is has been implicated in the pathogenesis of fibrosis in response to epithelial stress and injury in multiple organs, including the lung, kidney, liver, and peritoneum, the present data suggest that Gas6 could be used to develop preventive and therapeutic strategies to limit pathologic fibrosis in diverse organ diseases.

## Figures and Tables

**Figure 1 cells-08-00643-f001:**
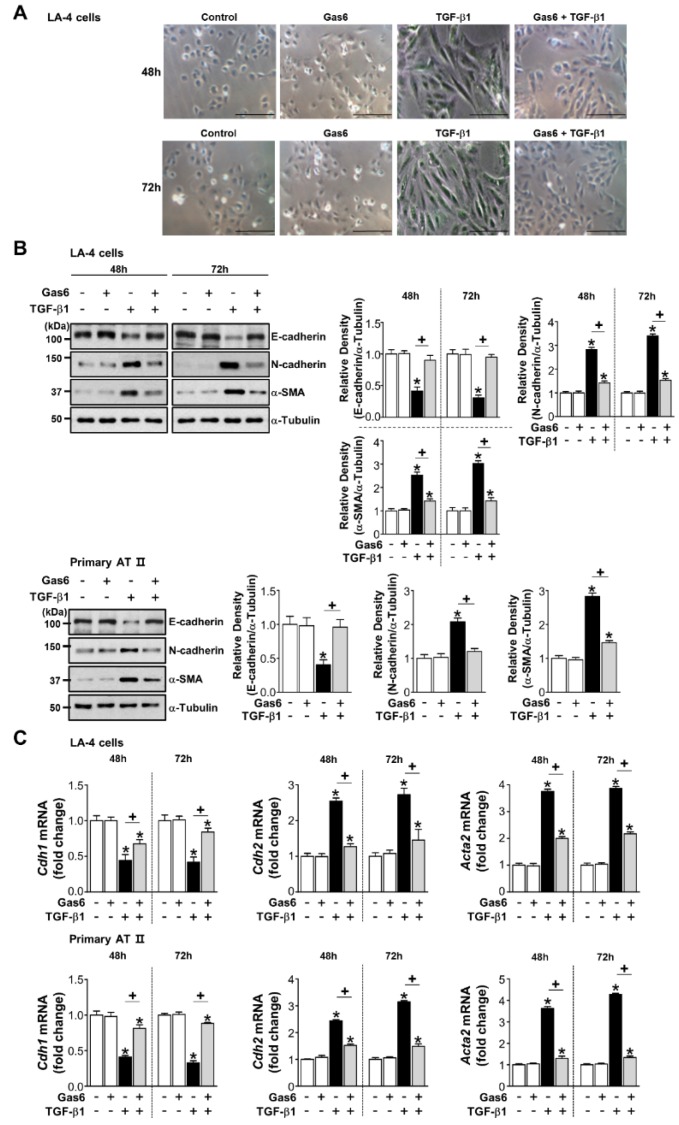
Growth arrest-specific protein 6 (Gas6) pretreatment inhibits transforming growth factor (TGF)-β1-induced epithelial-mesenchymal transition (EMT) in lung epithelial cells (ECs). (**A**–**C**) LA-4 and ATII ECs were pretreated with 400 ng/mL Gas6 for 20 h prior to 10 ng/mL TGF-β1 treatment for 48 or 72 h. (**A**) Morphological changes in LA-4 ECs were examined by phase-contrast microscopy. Scale bars = 50 μm. Results are representative of three independent experiments. (**B**) Immunoblots of total cell lysates were performed with anti-E-cadherin, -N-cadherin, or -α-SMA antibodies. Densitometry of the relative abundances of the indicated EMT markers. Alpha-tubulin was used as a control. (**C**) The amount of EMT markers’ mRNAs in cell lysates was analysed by real-time PCR and normalized to that of hypoxanthine phosphoribosyltransferase. Values represent the mean ± S.E. of three independent experiments. * *P* < 0.05; compared with control; ^+^
*P* < 0.05 as indicated.

**Figure 2 cells-08-00643-f002:**
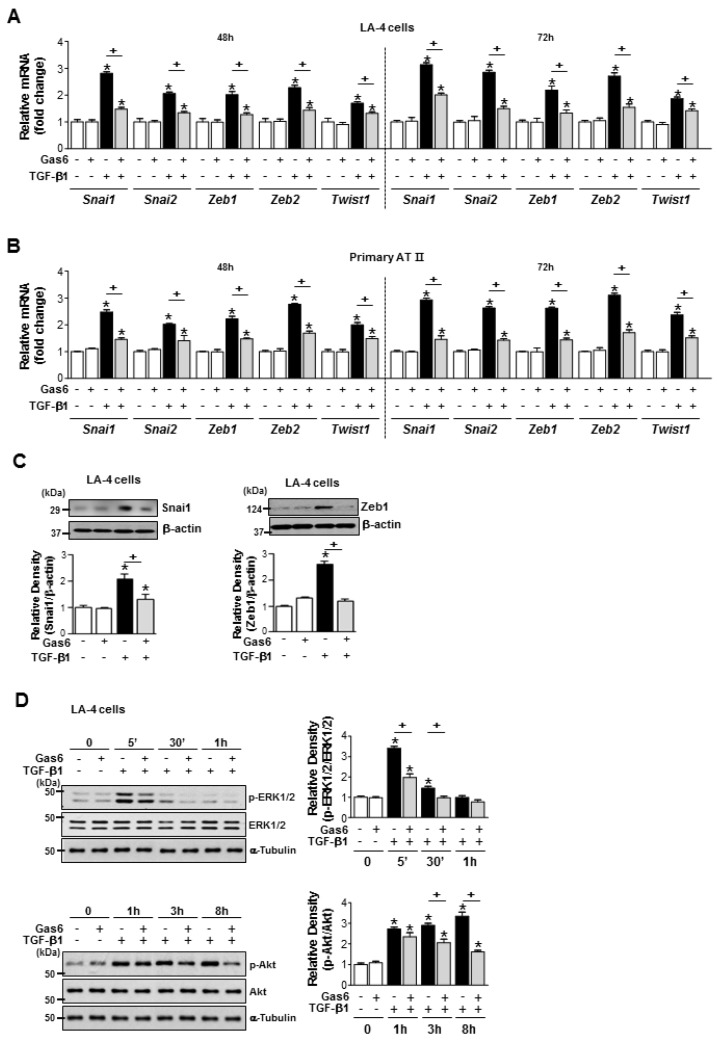
Growth arrest-specific protein 6 (Gas6) pretreatment reduces epithelial-mesenchymal transition (EMT)-regulating transcription factor expression and blocks Smad-independent transforming growth factor (TGF)-β1 signalling in epithelial cells. (**A**–**C**) LA-4 and ATII epithelial cells (ECs) were pretreated with 400 ng/mL Gas6 20 h prior to 10 ng/mL TGF-β1 stimulation for 48 or 72 h. (**A**,**B**) The amounts of *Snai1/2, Zeb1/2,* and *Twist1* mRNA were analysed by real-time PCR and normalized to that of hypoxanthine phosphoribosyltransferase (*Hprt*). (**C**,**D**) Representative immunoblots of LA-4 EC lysates were performed with anti-Snail1, -Zeb1, -total/phosphorylated ERK1/2, and -Akt protein antibodies. Beta-actin or alpha-tubulin was used as a loading control. Densitometric analysis of the indicated protein abundances. Data in all bar graphs are the mean ± S.E. of three independent experiments. * *P* < 0.05 compared with control; ^+^
*P* < 0.05 as indicated.

**Figure 3 cells-08-00643-f003:**
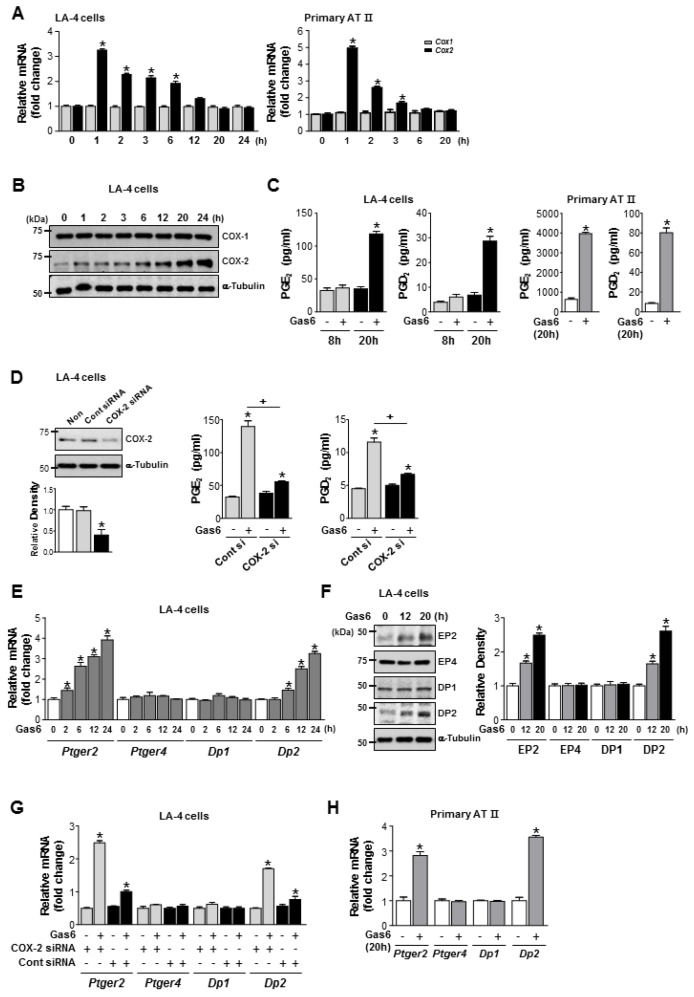
Cyclooxygenase (COX)-2 signaling is required for growth arrest-specific protein 6 (Gas6)-induced production of prostaglandin (PG)E_2_, PGD_2_, and their receptors. (**A**–**C**) LA-4 and primary alveolar type II (AT II) epithelial cells (ECs) were treated with 400 ng/mL Gas6 for the times indicated. (**A**) qPCR analysis of *Cox2* and *Cox1* mRNAs in cell lysates. (**B**) Representative immunoblots of LA-4 EC lysates were performed with anti-COX-2, -COX-1, or -α-tubulin antibodies. (**C**) PGE_2_ or PGD_2_ levels in conditioned media from LA-4 and AT II ECs were measured by enzyme immunoassay. (**D**) Immunoblots of total cell lysates were performed with anti-COX-2 antibodies in LA-4 ECs transfected with COX-2 specific or control siRNA for 6 h. Densitometric analysis of the COX-2 relative abundances. PGE_2_ and PGD_2_ levels in conditioned media from LA-4 ECs transfected with COX-2 specific or control siRNA for 6 h prior to treatment with 400 ng/mL Gas6 for 20 h were measured by EIA. (**E**) qPCR analysis of *Ptger2, Ptger4, Dp1,* and *Dp2* mRNA in LA-4 ECs treated with 400 ng/mL Gas6 for the time indicated. (**F**) Immunoblot analysis of EP2, EP4, DP1, or DP2 in LA-4 cells. Densitometric analysis of the indicated receptor’ relative abundances. (**G**) qPCR analysis of *Ptger2, Ptger4, Dp1,* and *Dp2* mRNA in LA-4 ECs transfected with COX-2 specific or control siRNA for 6 h prior to treatment with 400 ng/mL Gas6 for 20 h. (**H**) qPCR analysis of *Ptger2, Ptger4, Dp1,* and *Dp2* mRNA in ATII ECs treated with 400 ng/mL Gas6 for the time indicated. Data in all bar graphs are the mean ± S.E. of three independent experiments. * *P* < 0.05 compared with control; ^+^
*P* < 0.05 as indicated.

**Figure 4 cells-08-00643-f004:**
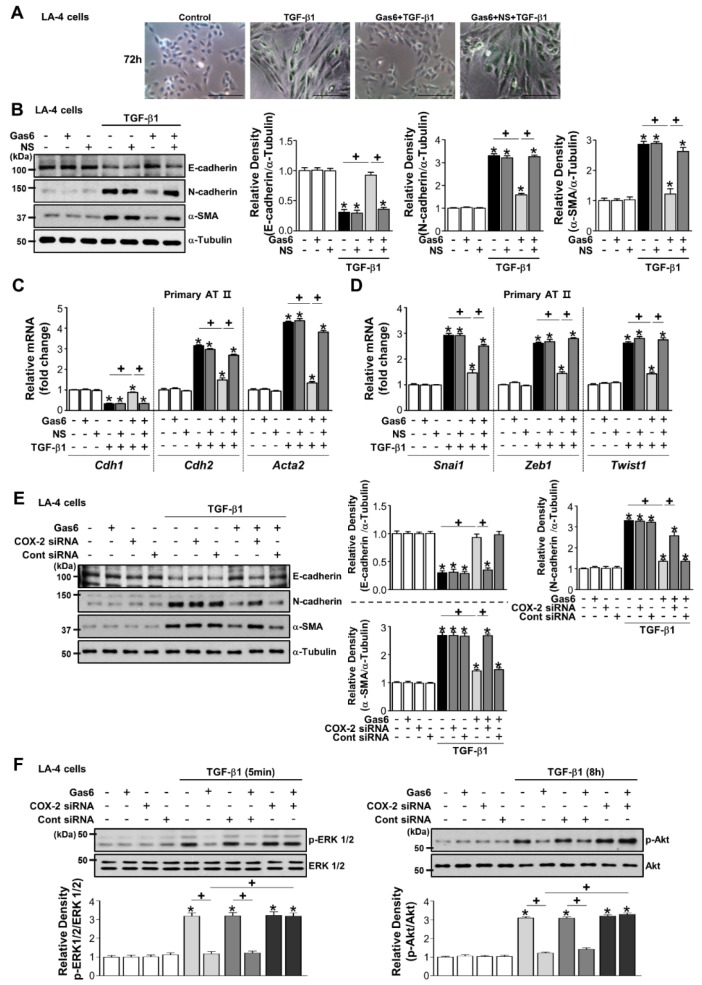
Cyclooxygenase (COX)-2-derived signaling mediates growth arrest-specific protein 6 (Gas6)-induced epithelial-mesenchymal transition (EMT) inhibition. (**A**–**D**) LA-4 ECs were pretreated with 10 μM NS-398 1 h before 400 ng/mL Gas6 treatment for 20 h and then stimulated with 10 ng/mL TGF-β1 treatment for 72 h. (**A**) Morphological changes in the cells were examined by phase-contrast microscopy. Scale bars = 50 μm. (**B**) Immunoblots of total cell lysates were performed with anti-E-cadherin, -N-cadherin, or -α-SMA antibodies. Densitometric analysis of the indicated EMT markers’ relative abundances. (**C**,**D**) Primary AT II cells were pretreated with 10 μM NS-398 1 h before 400 ng/mL Gas6 treatment for 20 h and then stimulated with 10 ng/mL TGF-β1 treatment for 72 h. qPCR analysis of the mRNAs of EMT markers and EMT-regulating transcription factors. (**E**,**F**) LA-4 ECs were transfected with COX-2-specific or control siRNAs for 6 h prior to treatment with 400 ng/mL Gas6 for 20 h and then stimulated with 10 ng/mL TGF-β1 for 72 h or the times indicated. Representative immunoblots of LA-4 EC lysates were performed with anti-E-cadherin, -N-cadherin, -α-SMA, -total/phosphorylated ERK1/2, and -Akt protein antibodies. Densitometric analysis of the indicated protein abundances. Data in all bar graphs are the mean ± S.E. of three independent experiments. * *P* < 0.05 compared with control; ^+^
*P* < 0.05 as indicated.

**Figure 5 cells-08-00643-f005:**
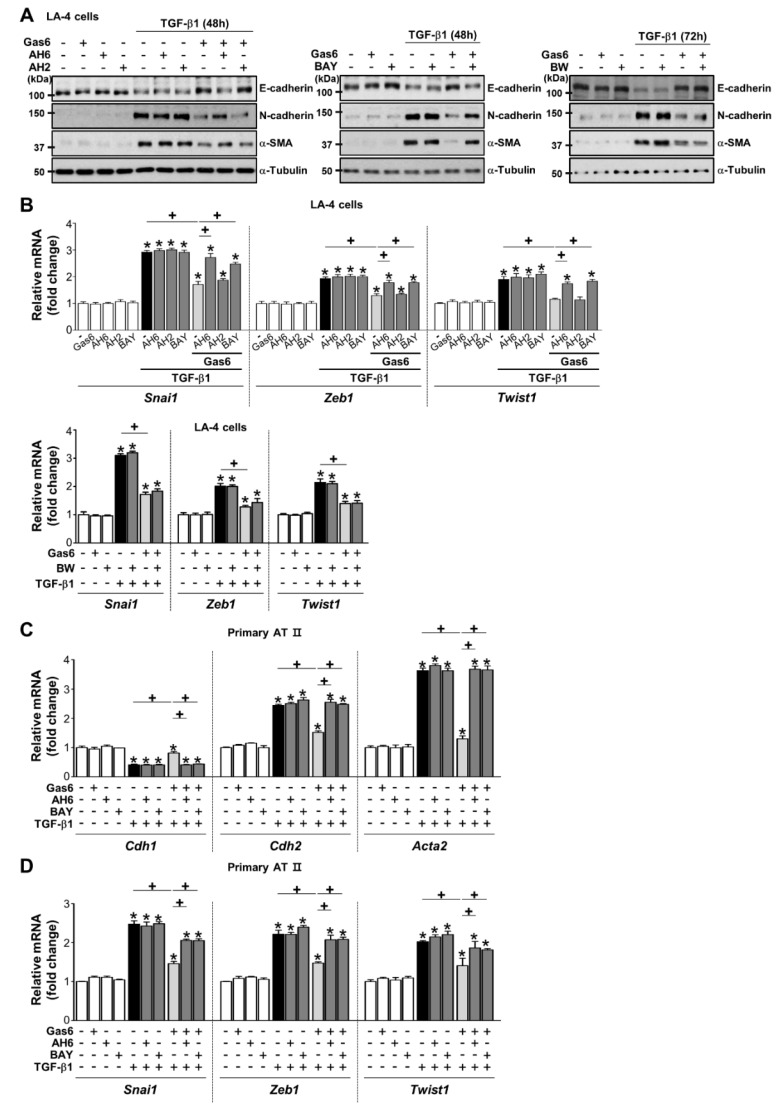
Prostaglandin (PG)E_2_ and PGD_2_ secretion inhibits growth arrest-specific protein 6 (Gas6)-induced epithelial-mesenchymal transition (EMT) via their receptors. (**A**–**D**) LA-4 and AT II epithelial cells (ECs) were stimulated with 400 ng/mL Gas6 for 20 h and then stimulated with 10 ng/mL TGF-β1 for 48 or 72 h with or without antagonists of EP2 (AH-6809), EP4 (AH-23848), DP1 (BW-A868C), or DP2 (BAY-u3405), each at a dose of 10 μM. (**A**) Representative immunoblots of total cell lysates were performed with anti-E-cadherin, -N-cadherin, or -α-SMA antibodies. (**B**) qPCR analysis of the mRNAs of EMT transcription factors in LA-4 ECs. (**C**,**D**) qPCR analysis of the mRNAs of EMT markers and EMT transcription factors in primary AT II ECs. Values represent the mean ± S.E. of three independent experiments. * *P* < 0.05 compared with control; ^+^
*P* < 0.05 as indicated.

**Figure 6 cells-08-00643-f006:**
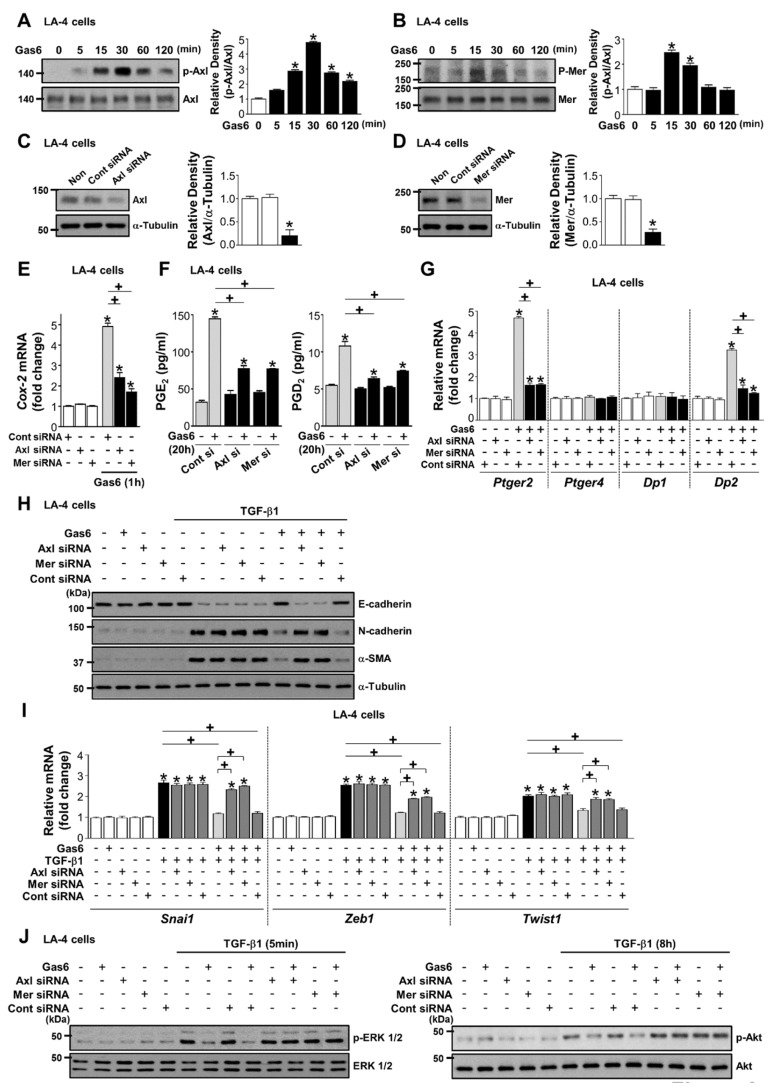
Activation of Axl or Mer mediates growth arrest-specific protein 6 (Gas6)-induced inhibition of COX-2 signaling and epithelial-mesenchymal transition (EMT) in LA-4 epithelial cells (ECs). (**A**,**B**) Immunoblot of total cell lysates were performed with anti-total/phosphorylated Axl and -Mer antibodies in LA-4 ECs treated with 400 ng/mL Gas6 for the times indicated. Densitometric analysis of the indicated protein abundances. (**C**,**D**) Immunoblots of total cell lysates were performed with anti-Axl, or -Mer antibodies in LA-4 ECs transfected with Axl, Mer, or control siRNA. Densitometric analysis of the indicated protein abundances. (**E**–**G**) LA-4 ECs were transfected with Axl, Mer, or control siRNAs for 48 h and then stimulated with 400 ng/mL Gas6. (**E**,**G**) qPCR analysis of the mRNAs of *Cox2, Ptger2, Ptger4, Dp1,* and *Dp2* in LA-4 EC lysates 1 or 20 h after Gas6 stimulation. (**F**) PGE_2_ and PGD_2_ levels in conditioned media 20 h after Gas6 stimulation were measured by enzyme immunoassay. (**H**–**J**) LA-4 ECs were transfected with Axl, Mer, or control siRNAs for 48 h prior to treatment with 400 ng/mL Gas6 for 20 h and then stimulated with 10 ng/mL TGF-β1 for 72 h or the times indicated. (**H**) Representative immunoblots of total cell lysates with anti-E-cadherin, -N-cadherin, or -α-SMA antibodies in the indicated samples. (**I**) qPCR analysis of the mRNAs of EMT transcription factors. (**J**) Representative immunoblots of total cell lysates with anti-total/phosphorylated ERK1/2 and -Akt protein antibodies. Values represent the mean ± S.E. of three independent experiments. * *P* < 0.05; compared with control; ^+^
*P* < 0.05 as indicated.

**Figure 7 cells-08-00643-f007:**
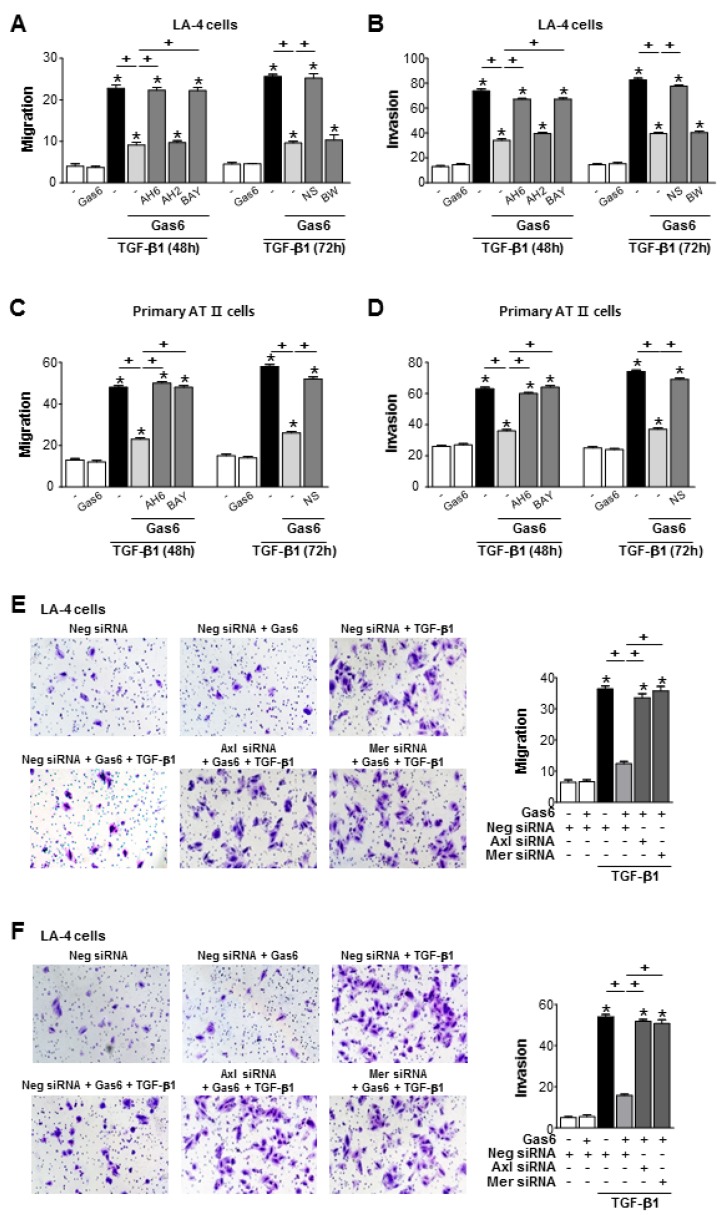
Growth arrest-specific protein 6 (Gas6)/Axl signaling inhibits migration and invasion of LA-4 and alveolar type II (AT II) epithelial cells (ECs) via prostaglandin (PG)E_2_ and PGD_2_. (**A**–**D**) LA-4 and primary AT II ECs were stimulated with 400 ng/mL Gas6 for 20 h and then stimulated with 10 ng/mL TGF-β1 with or without antagonists of EP2 (AH-6809), EP4 (AH-23848), DP1 (BW-A868C), or DP2 (BAY-u3405), each at a dose of 10 μM, for 48 or 72 h. The quantification of migrated or invaded cells in Boyden chambers. (**E**,**F**) LA-4 ECs were transfected with Axl, Mer, or control siRNAs for 48 h prior to treatment with 400 ng/mL Gas6 for 20 h and then stimulated with 10 ng/mL TGF-β1 for 72 h. The cells were visualized by phase-contrast microscopy for the analysis of migratory in (**E**) left and invasive in (**F**) left abilities using Fn-coated Transwell and Matrigel-coated Transwell plates, respectively. Scale bars: 100 μm. Quantification of cells that migrated in (**E**) right or invaded in (**F**) right. Values represent the mean ± S.E. of three independent experiments. * *P* < 0.05 compared with control; ^+^
*P* < 0.05 as indicated.

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
