# Peer review of "Gas6 Prevents Epithelial-Mesenchymal Transition in Alveolar Epithelial Cells via Production of PGE2, PGD2 and Their Receptors"

_cells, 2019, doi:10.3390/cells8070643_

Round 1

Reviewer 1 Report

The manuscript reports a study on the ability of Gas6 to prevent TGF-beta1 to induce EMT in murine epithelial cells. 

The issue is interesting and quite original, but there are some important concerns.

The first one is the design of the study that is quite troublesome to interpret.

The background of the study should be better explained because the current introduction does not provide enough explanation of the rationale of the study. I think that the Authors may be able to make an effort to improve the logical consequence of their reasoning about the background. Moreover, not all the readers may be familiar with the pathway Gas6-Tyro3/Axl/Mer and some more detail should be done in the Introduction and Discussion.

Also the aim of the study should be better stated in consequence of the background.

The results section is influenced by the lack of clarity in the introduction.

No image about the purity of primary ATII cells was provided.

Figure 2C is missing the loading control, please give an explanation about it.

The Discussion seems to be focused on cell migration and invasion, but there are no experiments clearly proving these issues. 

Another point to be addressed is the lacking of EMT markers. At least Zeb1-2 or Snail should be detected.

MInor point:

-line 313 Snail instead of Snai

Author Response

Reviewer #1:

The manuscript reports a study on the ability of Gas6 to prevent TGF-beta1 to induce EMT in murine epithelial cells. 

The issue is interesting and quite original, but there are some important concerns.

The first one is the design of the study that is quite troublesome to interpret.

The background of the study should be better explained because the current introduction does not provide enough explanation of the rationale of the study. I think that the Authors may be able to make an effort to improve the logical consequence of their reasoning about the background. Moreover, not all the readers may be familiar with the pathway Gas6-Tyro3/Axl/Mer and some more detail should be done in the Introduction and Discussion.

(Response 1) We appreciate the reviewer’s suggestion. We have made an effort to improve the logical consequence and added a detailed description regarding the pathway Gas6-Tyro3/Axl/Mer in details in Introduction and Discussion. Please refer lines from 30 to 76 in the Introduction and lines 335-338 in the Discussion.

Also the aim of the study should be better stated in consequence of the background.

(Response 2) We have stated the aim of the study in consequence of the background. Please refer lines from 64 to 76 in the Introduction.

The results section is influenced by the lack of clarity in the introduction.

(Response 3) As described in Response 1 and 2, we have added a detailed description in the Introduction.

No image about the purity of primary ATII cells was provided.

(Response 4) We previously reported representative confocal microscopic images of purified ATII cells (LSM5 PASCAL; Carl Zeiss, Jena, Germany) [Lee et al., Cell Death Dis. 2017, 8, e2860 ].

Supplementary Figure 4. Immunofluorescence staining for pro-SP-C in primary isolated alveolar type II cells. Immunofluorescence staining for pro-SP-C and nuclei (DAPI, blue) in the isolated alveolar type II cells from mice. Scale bars = 20 mm. Results are representative of three independent experiments.

Figure 2C is missing the loading control, please give an explanation about it.

(Response 5) We have added it in Figure 2C.

The Discussion seems to be focused on cell migration and invasion, but there are no experiments clearly proving these issues. 

(Response 6) We have investigated whether and how Gas6 inhibits TGF-b1-induced migration and invasion in LA-4 or ATII ECs. Please refer lines from 314 from 319 as well as Fig. 7 and Supplementary Fig. 6.

Another point to be addressed is the lacking of EMT markers. At least Zeb1-2 or Snail should be detected.

(Response 7) We have examined TGF-b1-induced changes in EMT markers such as decrease in E-cadherin expression, and increases in the expression of N-cadherin and α-smooth muscle actin (SMA). Moreover, we demonstrated the mRNA expression of EMT core regulators Snai1/2, Zeb1/2, and Twist in LA-4 and ATII ECs. We previously reported only mRNA expression of Snai1/2, Zeb1/2, and Twist in LA-4, ATII ECs, and cancer cell lines to prove anti-EMT effect by apoptotic cells (Kim et al., Cell Mol Immunol 2019 doi: 10.1038/s41423-019-0209-1. [Epub ahead of print]; Yoon et al., Sci Rep 2016, 6:20992.). In addition, others reported also mRNA expression of these transcription factors without their protein expression related EMT process (Gras et al., PLoS One. 2014, 9:e92254; Sarrio et al., Stem Cells 2012;30:292–303; Park et al., Genes Dev. 2008, 22:894–907, Gibbons et al., Genes Dev. 2009, 23:2140-2151, etc.).

Since the Editor of Cells ask us to submit the revised MS within 10 days, it is very hardly to get protein data in our revised MS until that day. If the data should be included, we need at least one month because we have to purchase newly the antibodies to detect Snai and/or Zeb protein expression.           

MInor point:

-line 313 Snail instead of Snai

(Response 8) As suggested, we have correctly described as Snail.

We very much appreciate the suggestions raised by the reviewer. We hope that our manuscript will now be accepted for publication in Cells.

Sincerely,

Jihee Lee Kang, MD, PhD

Professor

Department of Physiology

College of Medicine

Ewha Womans University

Reviewer 2 Report

The study by Jung et al describes the inhibition of epithelial-mesenchymal transition (EMT), a process which has been suggested to occur in lung fibrosis, by growth arrest-specific protein 6 (Gas6) in murine alveolar epithelial type-II cells in vitro, through induction of production of PGE2 and PGD2 as well as of their receptors.

The paper by Jung et al summarizes of a nicely done work. The manuscript is clearly written, and the experiments and figures are very well presented. Great paper!

The manuscript do only reveal some spelling mistakes and syntactic mistakes. The authors should carefully proof-read their manuscript, and perform in some instances a careful stylistic and syntactic revision to improve clarity of the manuscript.

Some examples for spelling errors:

1. Abstract, line 22: "PGD2" should be read as PGD2

2. Material and Methods, line 60: "phosphor-Smad2, phosphor-Smad3" should be read as "phospho-Smad2, phospho-Smad3"

Some examples for stylistic/syntactic errors:

1. Material and Methods, lines 83-86, the authors wrote: "After lavage of lungs with 1 ml saline, dispase (100 units) was instilled into cleared mice lungs, the lungs were incubated for 45 min at room temperature, lung tissue was separated from large bronchi by mechanical means, and tissue transferred to a Petri dish containing Dulbecco’s modified Eagle’s medium (DMEM) with 0.01% DNase I for 10 min at 37°C."

This sentence is too long, and its syntax is inadequate. Instead, two sentences should be written. Please write: "After lavage of lungs with 1 ml saline, dispase (100 units) was instilled into cleared mice lungs, followed by incubation of the lungs for 45 min at room temperature. Thereafter, the lung tissue was separated from large bronchi by mechanical means, and the tissues were transferred to a Petri dish containing Dulbecco’s modified Eagle’s medium (DMEM) with 0.01% DNase I, in which they were incubated for 10 min at 37°C."

Further, the gene names for mouse genes are not given correctly. Importantly, the correct gene names have to be italicized. And italicized correct gene names have to be given if the authors describe gene expression data in the manuscript text as well as in the legends to figures. Further, italicized correct gene names have to be given on the y-axes on respective graphs describing the gene expression. Italicized correct gene names should also be given in the Methods, when mentioning the gene-specific primers!

The mouse gene name for E-Cadherin is Cdh1, for N-Cadherin it is Cdh2, for α-SMA it is Acta2, for COX-1 it is Cox1, for Cox-2 it is Cox2, for EP2 it is Ptger2, for EP4 it is Ptger4, for DP1 it is Dp1 or Tfdp1, for DP2 it is Dp2 or Tfdp2, for HPRT it is Hprt.

Legend to Figure 6: Changing font-size in this figure-legend, the font-size should be uniformly.

.

Author Response

Reviewer #2:
Some examples for spelling errors:

1. Abstract, line 22: "PGD2" should be read as PGD2

(Response 1) We appreciate the reviewer’s suggestion. We have changed "PGD2" to "PGD2"

2. Material and Methods, line 60: "phosphor-Smad2, phosphor-Smad3" should be read as "phospho-Smad2, phospho-Smad3"

(Response 2) As suggested, we have corrected these.

Some examples for stylistic/syntactic errors:

1. Material and Methods, lines 83-86, the authors wrote: "After lavage of lungs with 1 ml saline, dispase (100 units) was instilled into cleared mice lungs, the lungs were incubated for 45 min at room temperature, lung tissue was separated from large bronchi by mechanical means, and tissue transferred to a Petri dish containing Dulbecco’s modified Eagle’s medium (DMEM) with 0.01% DNase I for 10 min at 37°C."

This sentence is too long, and its syntax is inadequate. Instead, two sentences should be written. Please write: "After lavage of lungs with 1 ml saline, dispase (100 units) was instilled into cleared mice lungs, followed by incubation of the lungs for 45 min at room temperature. Thereafter, the lung tissue was separated from large bronchi by mechanical means, and the tissues were transferred to a Petri dish containing Dulbecco’s modified Eagle’s medium (DMEM) with 0.01% DNase I, in which they were incubated for 10 min at 37°C."

(Response 1) We appreciate the reviewer’s suggestion. We have changed these.

Further, the gene names for mouse genes are not given correctly. Importantly, the correct gene names have to be italicized. And italicized correct gene names have to be given if the authors describe gene expression data in the manuscript text as well as in the legends to figures. Further, italicized correct gene names have to be given on the y-axes on respective graphs describing the gene expression. Italicized correct gene names should also be given in the Methods, when mentioning the gene-specific primers!

The mouse gene name for E-Cadherin is Cdh1, for N-Cadherin it is Cdh2, for α-SMA it is Acta2, for COX-1 it is Cox1, for Cox-2 it is Cox2, for EP2 it is Ptger2, for EP4 it is Ptger4, for DP1 it is Dp1 or Tfdp1, for DP2 it is Dp2 or Tfdp2, for HPRT it is Hprt.

(Response 2) As suggested, we have corrected gene names.

Legend to Figure 6: Changing font-size in this figure-legend, the font-size should be uniformly.

(Response 3) As suggested, we have corrected font-size in Figure 6.

We very much appreciate the suggestions raised by the reviewer. We hope that our manuscript will now be accepted for publication in Cells.

Sincerely,

Jihee Lee Kang, MD, PhD

Professor

Department of Physiology

College of Medicine

Ewha Womans University

Round 2

Reviewer 1 Report

The revised manuscript is improved Vs. previously submitted one.

I still believe that to enhance robustness of the study ut should be helpful to provide protein data. Thus, I suggest to detect Snai and Zeb1 protein expression to complete the study before publication.

Finally, these references should be added to the list:

- Salton F, Volpe MC, Confalonieri M. Epithelial-mesenchymal transition in the pathogenesis of idiopathic pulmonary fibrosis. Medicina 2019; 55: 83

- Espindola MS, Habiel DM, Narayanan R, Jones I, Coelho AL, Murray LA, Jiang D, Noble PW, Hogaboam CM. Targeting of TAM receptors ameliorate fibrotic mechanisms in idiopathic pulmonary fbrosis. Am J Respir Crit Care Med 2018; 197: 1443-1456.

Author Response

Comments and Suggestions for Authors

The revised manuscript is improved Vs. previously submitted one.

I still believe that to enhance robustness of the study ut should be helpful to provide protein data. Thus, I suggest to detect Snai and Zeb1 protein expression to complete the study before publication.

(Response ) As suggested, we have examined Snail1 and Zeb1 protein expression using Western blotting analysis. We have added the results in Fig. 2C.

Finally, these references should be added to the list:

- Salton F, Volpe MC, Confalonieri M. Epithelial-mesenchymal transition in the pathogenesis of idiopathic pulmonary fibrosis. Medicina 2019; 55: 83

- Espindola MS, Habiel DM, Narayanan R, Jones I, Coelho AL, Murray LA, Jiang D, Noble PW, Hogaboam CM. Targeting of TAM receptors ameliorate fibrotic mechanisms in idiopathic pulmonary fbrosis. Am J Respir Crit Care Med 2018; 197: 1443-1456.

(Response) As suggested, we have added these references. In addition, we have discussed about the controversial data related to the role of Gas6/Axl signaling and protein S, another TAM receptor ligand. Please refer lines from 404 to 410 in the Discussion.

We very much appreciate the suggestions raised by the reviewer. We hope that our manuscript will now be accepted for publication in Cells.

Sincerely,

Jihee Lee Kang, MD, PhD

Professor

Department of Physiology

College of Medicine

Ewha Womans University